

# The impact of suspended sediments on exchange flow in a macrotidal, hyperturbid estuary.

Cristian M. Rojas[1], Lauren Ross[1], Betty John Kaimathuruthy[2], Isabel Jalón-Rojas[2], Aldo Sottolichio[2], and Nicolas Huybrechts[3,4]

[1]Department of Civil and Environmental Engineering, University Of Maine, Maine, USA
[2]Univ. Bordeaux, CNRS, Bordeaux INP, EPOC, UMR 5805, F-33600, Pessac, France
[3]Cerema Risques, Eaux et Mer (CEREMA REM), RHITME Research Team, Margny-les-Compiegne F-60280, France
[4]Universite Rouen Normandie, Universite Caen Normandie, CNRS, Normandie Universite, M2C, UMR 6143, Margny Les Compiegne F-60280, France

**Correspondence:** Cristian M. Rojas (cristian.rojas@maine.edu)

**Abstract.** This study investigates the sensitivity of the total exchange flow (TEF) to the consideration of suspended sediment concentrations (SSC) in water density calculations. We consider two model scenarios of an idealized model based on a macroti­dal, highly turbid estuary: one where density depends on both salinity and SSC and another where density depends only on salinity. These models are used to understand how TEF volume inflow ($Q_{in}$) varies along the estuary over fortnightly and monthly time scales. The inflow TEF bulk-values between the two model scenarios show small differences (less than 4%) over 5 months. However, differences in TEF bulk volume inflow reach 7% downstream of the estuarine turbidity maximum (ETM), 22% at the ETM, and nearly 70% upstream of the ETM when quantified only over spring tides. The gradient Richardson number indicates that sediment-induced stratification suppress shear production $\sim$ 30 km further upstream when suspended sediments are an active tracer. The most profound impact on TEF occur during spring tides at -and upstream of- the ETM where sediment contributions to density exceed those from salinity. The sediment contribution leads to increased stratification in the upper-reaches of the estuary, effectively reducing the longitudinal density gradient and therefore the exchange flow as much as 2000 m$^3$ s$^{-1}$, indicating a dominant role of suspended sediments in attenuating exchange in hyperturbid estuaries. The results from this work highlight the importance of the consideration of SSC on density calculations on estuarine studies of the exchange flow.

## 1 Introduction

As a result of complex estuarine dynamics, estuaries trap sediments and develop convergent regions of suspended sediment concentration (SSC) known as estuarine turbidity maxima (ETM) (Burchard et al., 2018b). SSCs and ETMs are of interest because sediments impact morphology (Allen et al., 1980; Chernetsky et al., 2010), oxygen concentrations (Etcheber et al., 2007; de Jonge et al., 2014), primary production (Yoshiyama and Sharp, 2006), salinity intrusion (Zhu et al., 2021), and also have socioeconomic implications (Valle-Levinson, 2022) in estuaries and coastal waters.





Time-scales for fine sediment trapping can range from tidal to interannual (Jalón-Rojas et al., 2016; Ralston and Geyer, 2017), indicating that the amount of sediments accumulated in estuaries can be up to the equivalent of several years of fluvial sediment input (van Maanen and Sottolichio, 2018; Grasso et al., 2018; Ralston and Geyer, 2017). Over intratidal time scales, suspended sediments migrate upstream during flood tides and downstream during ebb tides which depends on the ebb-flood

asymmetry (Burchard et al., 2018b). Despite the large importance of tides, the tidal averaged location of ETMs is mainly related to the river flow, as high and low river discharge conditions cause ETMs to migrate downstream and upstream, respectively (Burchard et al., 2018b; Becker et al., 2018; van Maanen and Sottolichio, 2018; Grasso et al., 2018). The seasonal migration of ETMs varies from estuary to estuary, but are typically on the order of tens of kilometers. For example, the ETM location varies by more than 20 km in the Ems Estuary (de Jonge et al., 2014) and close to 70 km in the Gironde (Jalón-Rojas et al., 2015).

Nevertheless, irrespective of river conditions, spatially locked ETMs have also been reported in the Columbia River estuary (Hudson et al., 2017) or in the macrotidal Gironde Estuary where a second ETM have a more stable location (van Maanen and Sottolichio, 2018; Sottolichio and Castaing, 1999).

As sediments are flushed downstream by river outflows, without an opposite upstream transport, sediments would not accumulate and ETM formation would not be possible (van Maanen and Sottolichio, 2018; Zhu et al., 2021; Burchard et al., 2018b).

In tide-dominated estuaries, mechanisms such as tidal pumping (Becherer et al., 2016) or stokes drift (Díez-Minguito et al., 2014) could induce upstream net transport. However, the main mechanism is provided by the estuarine exchange flow which is typically characterized by a near-surface buoyant flow towards the ocean that is compensated by a near-bed landward flow of dense, salty water (Geyer and MacCready, 2014; MacCready and Geyer, 2010). This mechanism, also called gravitational circulation, provides the upstream transport needed to develop the ETM as higher concentration of sediments occurs near the

bottom (van Maanen and Sottolichio, 2018; Zhu et al., 2021). Since the estuarine exchange flow is an advective process that depends on the amount of mixing that occur in estuaries (Burchard et al., 2018a), it is sensitive to stratification, which can vary due to tidal processes. Mixing is affected by flood-ebb asymmetries causing enhanced turbulent mixing during flood tide and reducing it during ebb tide. This internal tidal asymmetry, also called tidal straining by some authors, cause enhanced stratification during ebb tide, also contributing to the estuarine circulation similarly to the gravitational circulation (Jay and

Musiak, 1994, 1996). The strain-induced periodic stratification (SIPS) also contributes to strength the estuarine exchange flow. During ebb, faster oceanward currents near the surface causes the straining of the density field which modifies the sheared velocity field. During flood, the straining decreases, the water column becomes more homogeneous and the near-bottom currents intensify (Simpson et al., 1990; Jay and Musiak, 1994).

The importance of stratification to exchange flows had led to extensive research focusing on the role of salinity-induced

stratification (see Geyer and MacCready, 2014; Valle-Levinson, 2021, and references therein). However, a much less studied effect is the induced stratification by suspended sediments. In a one-dimensional vertical numerical study, Winterwerp (2001) found that the decrease in the effective Von Kármán constant observed in laboratory experiments by Coleman (1981) can be explained by sediment-induced buoyancy effects. Further, they showed that even low SSCs (1 gL$^{-1}$) can affect turbulence fields, leading to the damping of mixing through sediment-induced stratification (Geyer, 1993; Zhu et al., 2021). In the case of

hyperturbid estuaries, where SSCs can reach $> 10$gL$^{-1}$ (Uncles et al., 2002), density can be controlled by a combination of




salinity and SSCs. An observational study in the Ems Estuary showed that the density structure near the ETM was controlled mainly by SSC reaching values up to 50 $gL^{-1}$ (Becker et al., 2018). These fluid mud conditions induced stratification that had an intratidal asymmetry. More recently, Bailey et al. (2024) showed that these effects are relevant even under lower SSC conditions (∼1-10 $gL^{-1}$), such as those found downstream of the ETM in the Ems. They found that a combination of sediment
and salinity-induced stratification dampened turbulent mixing when the downstream boundary of the ETM was approximately 5 km from their measurement station. However, they were not able to determine the spatial extent of the sediment-induced stratification along-channel or the effects on exchange flows due to the limited spatial and temporal coverage of their measurements.

Although it is clear that in turbid estuaries SSC can impact stratification, only a few studies have focused on the specific
impact on exchange flow. Using an analytical model of the tidally averaged flow in the Ems Estuary, Talke et al. (2009) studied the role of along-channel variability in SSC in generating longitudinal sediment gradients. These gradients were found to decrease the classical estuarine residual flow downstream of the ETM but strengthened it upstream (Talke et al., 2009). However, their analytical model was highly idealized using a constant eddy viscosity and assuming vertically well-mixed conditions in a simple domain, leaving open the question of whether these results extend to more realistic systems. More
recently, Zhu et al. (2021) used a 3D model to study in the Yangtze estuary confirming Talke's results showing that baroclinic circulation was enhanced upstream of the ETM when sediments and salinity were considered in the pressure gradient. The authors showed that the increase and decrease of the classical estuarine residual flow upstream and downstream of the ETM, respectively, were driven by longitudinal sediment-induced density gradients that cause divergence of sediment. However, this work was mostly focused on how sediment-induced density gradient influence the location of the ETM and, while the authors
calculated yearly average residual currents, a more thorough focus on the specific impact of exchange flow is still needed.

The present work built on considerations of these previous studies seeking to understand how the inclusion of suspended sediments in water density calculations affects the exchange flow using the total exchange flow (TEF) framework. The goals of this work are (1) to determine when and where density is impacted by SSC and (2) to determine how this impact influences the total exchange flow (TEF). We use idealized numerical simulations based on the hyperturbid and macrotidal Gironde Estuary
that incorporated sediment dynamics. The remainder of this study will present a description of the study region in section 2 and the implementation of the model and methodology in section 3. The results are highlighted in section 4 followed by a discussion in section 5. A summary is finally highlighted in section 6.

## 2   Study region

The Gironde is located on the southwest coast of France (Figure 1), covering an area of 635 $km^2$ (Jalón-Rojas et al., 2015)
and is one of the largest estuaries in Western Europe. Is ∼75 km funnel shape long from the estuary mouth to Ambes where the Garonne and Dordogne Rivers converge. The main navigational channel extends from the estuary mouth up to Bordeaux (Figure 1). The Garonne and Dordogne Rivers are two tributaries that drain into the estuary converging near the town of Ambes (red dot, Figure 1) with a combined discharge that varies between 50 and 2000 $m^3$ $s^{-1}$ with the seasons, and flood events that





reach peak discharge values of 5000 m$^3$ s$^{-1}$. The tidal intrusion length is 180 km (Bonneton et al., 2015) whereas the salt

intrusion length only reaches approximately 100 km upstream (Schmidt, 2020). The estuary is classified as macrotidal since it features semi-diurnal tides at the mouth with a range between 2.5 to 5 m within a neap-spring cycle (Ross and Sottolichio, 2016). Further, the funnel-like shape of the estuary causes the tidal wave and currents to be amplified as they propagate upstream, becoming highly asymmetric and flood dominant due to the amplification of both compound tides and overtides (Ross and Sottolichio, 2016).

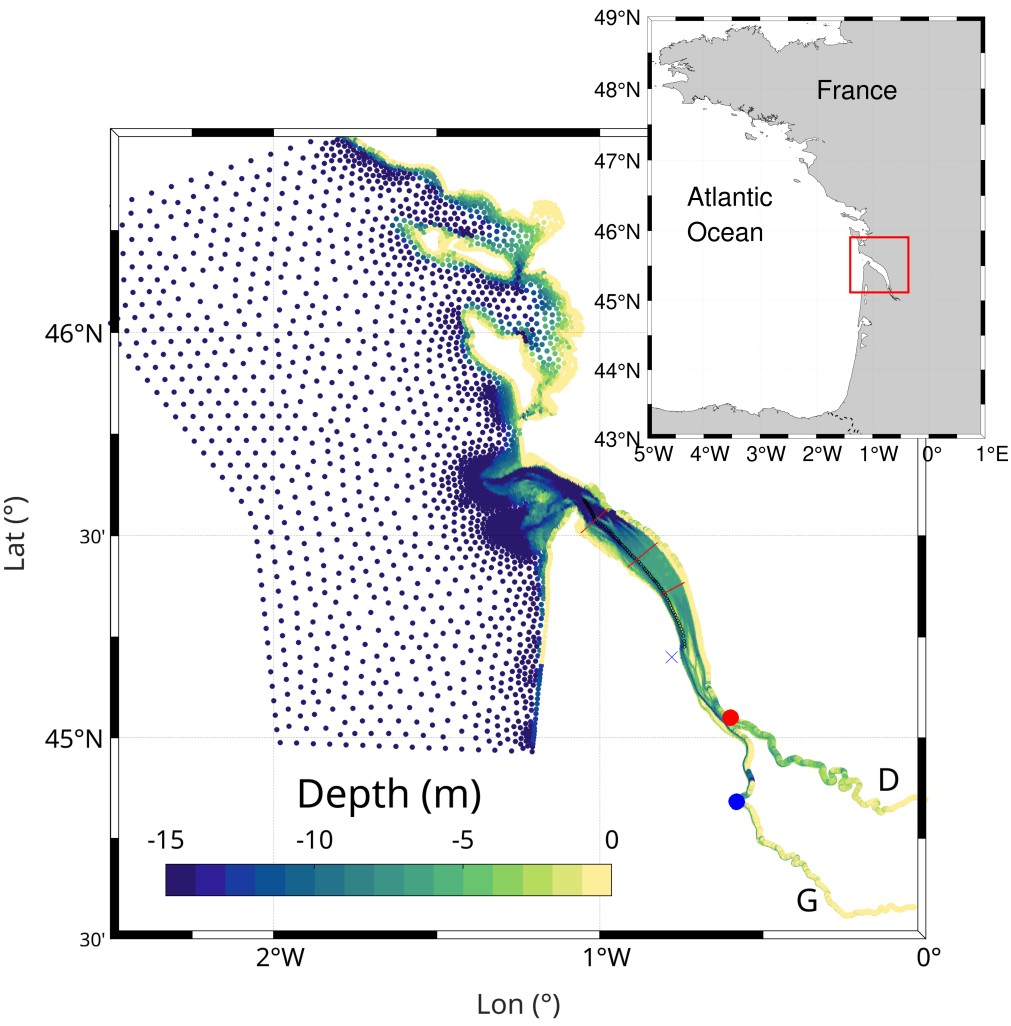

**Figure 1.** The Gironde estuary. The unstructured model grid is shown along with the location of the estuary in relation to the Atlantic coast of France (top right inset). The color represents the bathymetry (m) of the region, the red dot depicts the river confluence at Ambes, the blue dot the city of Bordeaux, the blue cross the city of Pauillac, and G and D represents the Garonne and Dordogne rivers, respectively. The black line depicts the navigational channel. The three red lines represents the location focused on the following sections.





The Gironde is considered a hyperturbid estuary with SSCs reaching $>10gL^{-1}$ (Defontaine et al., 2024) making this estuary a natural laboratory for investigating sediment transport dynamics(Allen et al., 1980; Jalón-Rojas et al., 2021; Sottolichio et al., 2000). As is the case with most turbid estuaries, the ETM location in the Gironde is dictated by the river regime (Jalón-Rojas et al., 2015). Seasonal river changes cause the location of the ETM to vary between 30-70 km in this system (Allen et al., 1980; van Maanen and Sottolichio, 2018). During high river discharge, the ETM is advected downstream (van Maanen and

Sottolichio, 2018), but during low river discharge periods, the ETM is formed landward of the salinity intrusion limit, well into the freshwater region (Allen et al., 1980). This is due to weak longitudinal salinity gradients allowing tidal processes, such as tidal pumping, to take a more prominent role in sediment transport dynamics (Allen et al., 1980). A previous study in the Gironde identified that the average flow produced by tidal asymmetry between ebb and flood was the main mechanism driving upstream transport of sediments and that balanced the downstream transport promoted by the river in this macrotidal estuary

(Allen et al., 1980).

## 3  Methods

### 3.1  Numerical model setup

A numerical model was implemented using the Telemac-3D finite element open-source model (Hervouet, 2007; Moulinec et al., 2011), following the configurations of Do et al. (2025). The model uses Prandtl's mixing length model with Munk Anderson

damping function for the vertical turbulence closure and constant values for horizontal turbulence (Hervouet, 2007). Wetting and drying options were activated to account for the tidal flats in the estuary (Figure 1). The model consists of nine equidistant terrain-following vertical layers that expand and contract following the tides with finer resolution near the bed. The model time-step was 30 seconds and snapshots of the model were saved every 20 minutes for 10 months of 2018 (February - November). Two upstream freshwater boundaries were defined and forced with realistic river discharge data available every 12 hours which

was collected by French government agencies (data available at data.eaufrance.fr). The salinity at the offshore boundary was set to 35 g kg$^{-1}$ based on previous work on the region (Alahmed et al., 2021). The tidal forcing was extracted from the North East Atlantic tidal model (Pairaud et al., 2008) including 46 tidal constituents and storm surges at the open offshore boundary. To study sediment dynamics, Telemac-3D was coupled with GAIA, a module of Telemac that simulates sediment transport for multiple sediment classes, bed evolution processes and that includes fine and cohesive suspended sediments that are actively

incorporated into hydrodynamic calculations (Tassi et al., 2023). In the present study, however, only one class of fine sediment is considered, and bed evolution is supposed to be negligible at the simulated time-scales.

In the configuration analyzed by Do et al. (2025), several settling-velocity parameterizations were tested in the Gironde estuary, each with advantages and limitations for reproducing SSC observations and ETM dynamics. For the present study, we adopt the van Leussen parameterization, which provided the best validation of SSC maxima. The present model only considers

a single cohesive sediment class, with a settling velocity computed dynamically through the Van Leussen flocculation parame-terization. While this formulation has clear advantages for reproducing the upstream migration of the ETM and capturing high SSC levels (Do et al., 2025), it also tends to underestimate SSC near the bottom during neap tides. Nevertheless, the focus of





this work is to study whether SSC could impact the exchange flow and because the original configuration was already validated, we consider the model suitable to study the impact of suspended sediments on the exchange flow.

## 3.2 Quantification of water Density - With & without suspended sediments

Several studies have shown that suspended sediments have an impact on hydrodynamics (Lu et al., 2020; Becker et al., 2018; Bailey et al., 2024), tidal amplification (Zhu et al., 2021), velocity shear (Winterwerp, 2001), stratification (Zhu et al., 2021; Lu et al., 2020; Geyer, 1993) or lateral circulation (Zhou and Tang, 2025). Therefore, to isolate sediment-induced effects on the exchange flow we ran two models: one where water density is quantified using both SSC and salinity, what we refer to as the "active model" ($\rho_{all}$), and another where water density is quantified using only salinity, referred to as the "passive model " ($\rho_{Sal}$). Both models used the same GAIA configuration to model sediment transport.

To differentiate the effects of salinity and SSC on water density, we assume that the total density ($\rho_{all}$) can be decomposed considering the contribution by salinity ($\rho_{Sal}$) and suspended sediments ($\rho_{SSC}$) independently following previous works (van Maanen and Sottolichio, 2018; Talke et al., 2009; Guan et al., 2005). Therefore, the water density can be decomposed as:

$$\rho_{all} = \rho_{sal} + \rho_{ssc}. \tag{1}$$

Telemac-3D computes the variation of density due to salinity as:

$$\rho_{Sal} = \rho_{ref} + 0.75 S, \tag{2}$$

where, $\rho_{ref} = 999.972$ kg m$^{-3}$ is the reference density at 4°C when the salinity $S$ is zero. The variation of density considering the sediment concentration is:

$$\rho_{SSC} = SSC \times \left( \frac{\rho_s - \rho_{ref}}{\rho_s} \right) = 0.62 SSC, \tag{3}$$

where $\rho_s$ is the specific density for primary particles of cohesive sediment and $SSC$ is the suspended sediment concentration. We used $\rho_s = 2650$ kg m$^{-3}$ following the value used in Do et al. (2025). Both models include suspended sediments but only in the active model they are considered into water density calculations. The active model corresponds to the RY3 configuration model by Do et al. (2025). An analysis of the passive model showed that the density of the model $\rho_{all}$ and the density due to only salinity $\rho_{Sal}$ have differences of the order of $1 \times 10^{-7}$ kg m$^{-3}$, which are likely due to numerical errors. Temperature effects were neglected into density calculations in both models, which is typical in estuarine studies.

The influence of suspended sediment on stratification was analyzed considering the gradient Richardson number $Ri = N^2/S^2$, a dimensionless ratio between buoyancy and velocity shear. The buoyancy frequency, $N^2$, was calculated as

$$N^2 = -\frac{g}{\rho_0} \frac{\partial \rho}{\partial z}, \tag{4}$$

where $g$ is acceleration due to gravity and $\rho_0$ is the mean density. The squared vertical shear $S^2$ was calculated as

$$S^2 = \left( \frac{\partial u}{\partial z} \right)^2 + \left( \frac{\partial v}{\partial z} \right)^2. \tag{5}$$





We will denote the buoyancy frequency and the gradient Richardson number as $N^2_{all}$ and $Ri_{all}$, respectively, to indicate the use of $\rho_{all}$ which includes both SSC and salinity. Similarly, we will denote as $N_{Sal}$ and $Ri_{Sal}$ to indicate the use of $\rho_{Sal}$ in the passive model. The value of $Ri > 0.25$ is usually considered indicative of total suppression of shear-driving mixing due to density changes (Miles, 1961). To differentiate the effects of salinity and suspended sediments on the vertical density gradient, a linear decomposition was calculated as:

$$\frac{\partial \rho_{all}}{\partial z} = \frac{\partial \rho_{SSC}}{\partial z} + \frac{\partial \rho_{Sal}}{\partial z}. \tag{6}$$

To determine the dominant periods of variability of the vertical density gradients presented above, and to establish links between stratification induced by salinity and SSC on the exchange flow, we applied a wavelet analysis to the time series of each density gradient component. This includes the total density gradient ($\partial \rho_{all}/\partial z$), the salinity-induced density gradient ($\partial \rho_{Sal}/\partial z$), and the sediment component ($\partial \rho_{SSC}/\partial z$) according to Eq. (6) (Torrence and Compo, 1998). Because the wavelet formulation developed by Torrence and Compo (1998) has a bias in the spectral estimate favoring large-scale features, we applied the wavelet rectification algorithm of Liu et al. (2007) allowing the comparison of the spectral peaks across scales.

### 3.3 Total Exchange Flow

The total exchange flow (TEF) analysis framework is a consistent way to calculate water and salt fluxes in estuarine systems using salinity coordinates according to the Knudsen relations (MacCready, 2011; Knudsen, 1900). In this study we follow the dividing salinity methodology where the TEF analysis framework was generalized to scenarios where the salinity ranges through a cross-section can be small (Lorenz et al., 2019; MacCready et al., 2018), as can be the case in the Gironde Estuary. The time-averaged volume transport ($Q^c$) of a tracer $c$, through a cross-sectional area $A$, is defined as:

$$Q^c(S) = \left\langle \int_{A(S)} c u \, dA \right\rangle, \tag{7}$$

where $u$ is the velocity component normal to the cross-section and $\langle \rangle$ represents a temporal average. The integration in (7) is carried out in the area with salinity greater than $S$. The convention is that a positive along-channel velocity, $u$, is directed in-estuary. The volume flux of the tracer $c$ per salinity class is obtained by differentiating $Q^c_S$ with respect to $S$:

$$q^c(S) = -\frac{\partial Q^c(S)}{\partial S}. \tag{8}$$

The inflow and outflow components of the volume transport, or bulk values, can be obtained using no tracer, ($c = 1$), as:

$$Q_{in} = \int_{S_{div}}^{S_{max}} q \, dS, \quad Q_{out} = \int_{S_{min}}^{S_{div}} q \, dS, \tag{9}$$

where the integrals are carried out using the extrema of $Q(S)$ to define the salinity class $S_{div}$ that separate inflows and outflows. The parameters $S_{min}$ and $S_{max}$ represent the minimum and maximum salinity that occurs in the section, respectively. The





minimum and maximum salinity values, as well as the dividing salinity classes, were calculated following the algorithm

developed by Lorenz et al. (2019).

The TEF volume transports were computed using both the active ($Q_{in}^{\text{Active}}$ and $Q_{out}^{\text{Active}}$) and the passive ($Q_{in}^{\text{Passive}}$ and $Q_{out}^{\text{Passive}}$) models. Calculations of $Q_{in} + Q_{out}$ were carried out in both models to check that mass was always conserved (MacCready, 2011). All time-series of TEF, SSCs, gradient Richardson number, and vertical density gradients were filtered using a 25-hour Godin filter to remove tidal variability (Thomson and Emery, 2014). To analyze the differences between TEF volume inflows

quantified with the active and passive models we applied a subtraction as,

$$Q_{in}^{\text{Passive-Active}} = Q_{in}^{\text{Passive}} - Q_{in}^{\text{Active}}. \tag{10}$$

To assess the longitudinal variability of $Q_{in}^{\text{Passive}}$ versus $Q_{in}^{\text{Active}}$, three along-channel sections were considered in this study according to their location relative to the ETM (red lines, Figure 1). The middle location was chosen to be the location where the ETM is developed during the wet period while the other two were downstream and upstream of the ETM. We also computed

the bulk-values of the volume transports for spring and neap tides and also over a 5-month period (February-June).

Since Telemac-3D uses an unstructured grid, pre-processing was necessary before any analysis. An interpolation of current velocities, water levels, salinity, and sediment onto a uniform grid was required before TEF quantification. The interpolation was carried out such that the grid resolution was close to the minimum distance between adjacent points of the numerical grid. The velocities were rotated to a normal axis with respect to the channel orientation using a principal-axis analysis resulting in

cross-sectionally and tidally averaged secondary flows of zero, ensuring mass conservation (Rhoads and Kenworthy, 1998).

Finally, we also calculated the along-channel density gradient, $\partial\rho/\partial x$, to quantify the change in the baroclinic forcing due to sediments. For this, we considered the cross-sectional average density the mouth and the head of the estuary. We then applied a 25-hour Godin filter to remove tidal variability (Thomson and Emery, 2014).

## 4 Results

### 4.1 The ETM and the impact of SSC over water density

To study the importance of the SSC over density calculations, the following sections are focused on the wet-season period (February-June). During this period, the ETM is located in the downstream region of the estuary and it well reproduced by the model (Figure 2c). It span $\sim 70$ km with maximum depth-averaged concentrations close to 3 gL$^{-1}$. Particularly, enhanced SSC exceeding depth averages of 4 gL$^{-1}$ were found after events of increased river discharge (exceeding 1500 m$^3$ s$^{-1}$) in February,

April, or June but only during spring tides.

The depth-averaged SSC shows a fortnightly cycle with enhanced concentrations during spring tides, reaching up to 3 gL$^{-1}$, and decreased values during neap tides, with concentrations less than 0.2 gL$^{-1}$. The concentration levels during spring tides are in agreement with those found in the Gironde indicating that the model is adequately reproducing the ETM. Lower depth-averaged SSC during neap tides are also expected as during these periods, less energetic currents allow the settlement

of suspended sediments. However, our model indicates unrealistic low values near the bottom (<0.5 gL−1) compared to





previously reported concentrations (Sottolichio et al., 2000; Sottolichio and Castaing, 1999). Since the focus of this work is not on SSC dynamics, we consider the results sufficiently reasonable to study the effects of SSC on water density calculations. Further, the selected settling velocity approach maintain the location of the ETM (Figure 2c) which also allow us to study the effect of SSC on monthly time-scales.

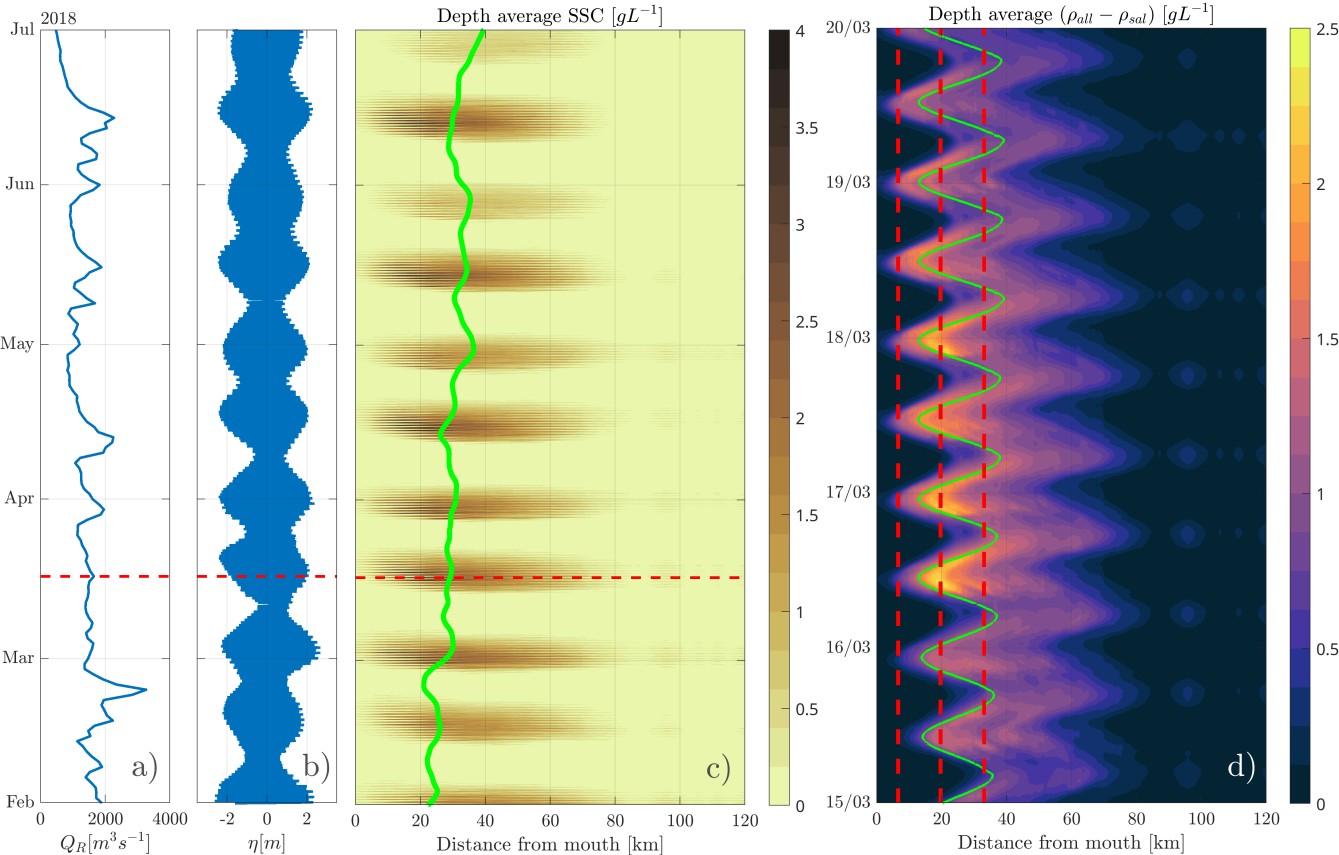

**Figure 2.** Forcing and sediment availability. Panels (a) and (b) are the combined river discharge and the tidal range at the mouth, respectively. Panel (c) is the depth averaged SSC along the estuary and panel (d) is the depth average of the difference $(\rho_{all} - \rho_{Sal})$. Panel (c) corresponds to the active model whereas panel (d) correspond to the difference between the active and the passive model. The horizontal red line in (a), (b) and (c) correspond to the period shown in panel (d). The filtered and unfiltered location of the 2 g kg$^{-1}$ isohaline at the bottom is depicted with a green line in panels (c) and (d), respectively. The vertical red dashed lines in (d) depict the along-channel locations of the three transects considered in the remainder of this study.

To examine where SSC have a large impact over density calculations, we calculate the depth-average of the difference between the active model ($\rho_{all}$) and the passive model ($\rho_{sal}$, Figure 2d). On intratidal time-scales, as SSC are transported upstream during flood and downstream during ebb. The differences span a large region covering ∼50 km during flooding and ∼40 km during ebbing. The differences increases with the tidal range, exceeding 2 gL$^{-1}$ when the tidal range is maximum (>4





m). During these periods, the differences between $\rho_{all}$ and $\rho_{Sal}$ are maximum near the 2 g kg$^{-1}$ isohaline (Figure 2d). Based

on the location of this maximum difference, we define three estuarine cross-sections: downstream (6.5 km from the mouth), ETM (19.6 km), and upstream (33 km), represented by vertical dashed-red lines in Figure 2d. In the following section, we calculate TEF quantities focusing on these three cross-sections.

## 4.2    Effects of SSC on TEF

To study the effects of SSC on exchange flow, time-series of TEF volume transport (Eq. 9) were calculated for both the active

and passive models at three cross-channel locations (red lines in Figure 1). For simplicity, we focus only on the TEF inflow $Q_{in}$ for the presentation of the results (Figure 3). We also calculated the longitudinal density gradient for both models.

The longitudinal density gradient appears qualitatively to be influenced by the river discharge regime. As the river discharge decreases, the longitudinal density gradient from both the active and passive models was enhanced. The combined river regime during February varied between 1500 to 2000 m$^3$s$^{-1}$ and the longitudinal density gradient varied between 0.18 to 0.25 kg m$^{-3}$

235    per km. Towards the end of June, when the combined river discharge was below 1000 m$^3$ s$^{-1}$, the longitudinal density gradient was $> 0.3$ kg m$^{-3}$ per km.

In general, the TEF volume inflow $Q_{in}$ was larger at the mouth of the estuary (downstream location) and continually decreased towards the head (ETM and upstream locations, Figure 3). The volume inflow $Q_{in}$ at the downstream and the ETM sections exhibit fortnightly variability with largest values during spring tides, which is enhanced at the downstream location.

At this location, the inflow from both the active and passive models peaks approximately 5000 m$^3$ s$^{-1}$ during spring tides and dips below 2000 m$^3$ s$^{-1}$ during neap tides (Figure 3b). Similar variability occurred at the ETM location in both the active and passive models, but with lower magnitude and with other higher frequency variability (around 7 days) also detectable (Figure 3c). In the case of the upstream section, there were instances during neap tides (i.e., 02/11 and 02/25) that the volume $Q_{in}$ decrease near null values, likely due to the lack of salinity at those times due to increased river discharge (Figure 3a,d).

The differences between the TEF volume inflow for the active ($Q_{in}^{\text{Active}}$) and passive ($Q_{in}^{\text{Passive}}$) models were largest during spring tides when SSCs reached their highest values. During spring tides, the longitudinal density gradient decreased in the active model compared to the passive model, although slightly. During these periods, peak differences up to $\sim 1000$ m$^3$ s$^{-1}$ occurred downstream, $\sim 2000$ m$^3$ s$^{-1}$ at the ETM and $\sim 1000$ m$^3$ s$^{-1}$ upstream (Figure 3b,c,d). The differences between models was smaller when SSCs decrease during neap tides due to large settlement of sediments. During periods of low SSC

(neap tides), the TEF volume inflow $Q_{in}^{\text{Active}}$ and $Q_{in}^{\text{Passive}}$ have negligible differences of the order of $\mathcal{O}(10^{-7})$ m$^3$ s$^{-1}$ in all three sections (Figure 3b,c,d). Similarly, the longitudinal density gradient difference between models was $< 10^{-4}$ kg m$^{-3}$ per km during neap tides.

In the downstream section during spring tides, $Q_{in}^{\text{Active}}$ was at times smaller and at other times larger than $Q_{in}^{\text{Passive}}$, indicating that including SSC in density can either decrease or increase the TEF volume inflow (Figure 3). However, in each case, the

differences were not as pronounced as those at the stations further upstream. At the ETM and upstream section, the overall effect of SSC led to a decrease in the TEF volume inflow ($Q_{in}^{\text{Active}} < Q_{in}^{\text{Passive}}$), showing that TEF volume inflows can be reduced by over 2 times when sediments are considered in density calculations. When considering the entire period shown in Figure 3





(5 months), the TEF volume inflow bulk-values did not change greatly between the active and passive models (see Table 1). The percentage difference was less than 4% downstream, 1.35% at the ETM and 2.1% in the upstream section. However, the

difference between the TEF volume inflow $Q_{in}$ along the estuary when considering neap and spring tides increased landward suggesting a more important role of SSCs upstream. Noticeably, the mean percentage difference of the volume inflow was more than $\sim$22% at the ETM location and more than $\sim$70% at the upstream section, suggesting that SSCs exert a larger influence on exchange flows at the ETM and upstream.

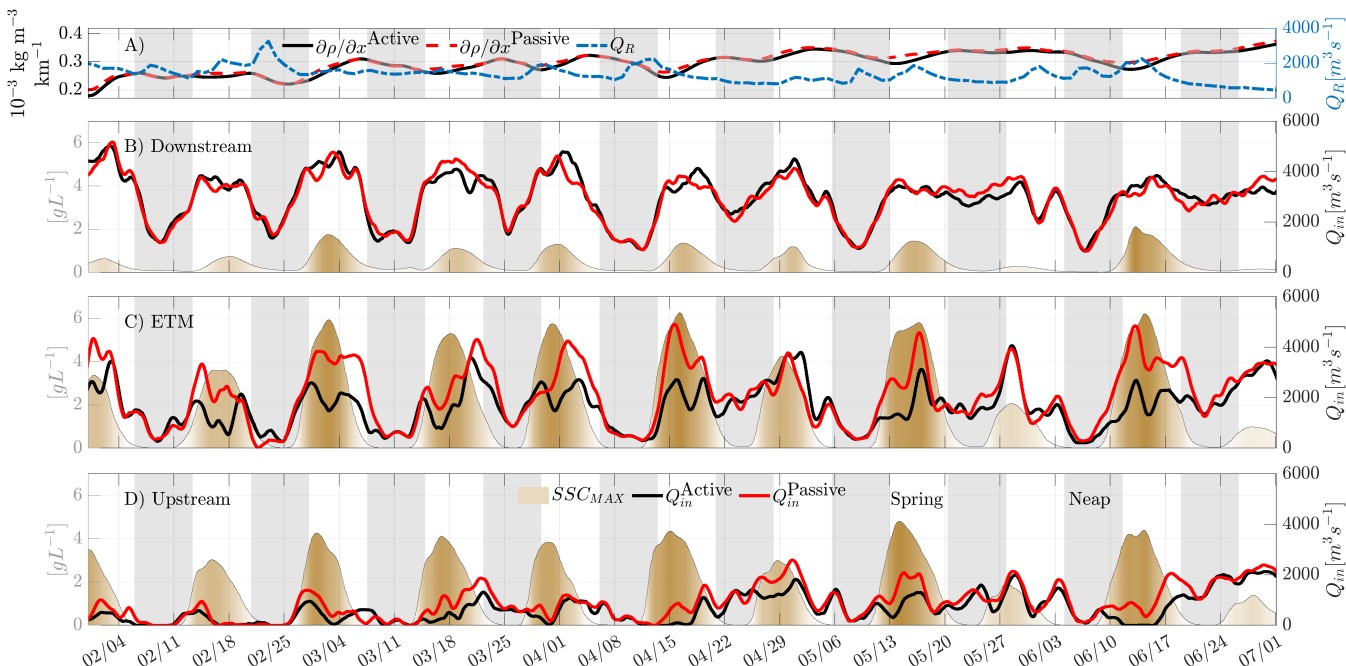

**Figure 3.** The combined river discharge from both the Garonne and Dordogne (blue line) together with the longitudinal density gradient ($\partial\rho/\partial x$) from the passive (black line) and the active (red-dashed line) model are shown in panel (A). Panel (B) shows the downstream, (C) ETM and (D) upstream section. The black and red lines in (B), (C) and (D) correspond to the TEF volume inflow from the active model $Q_{in}^{\text{Active}}$ and the TEF volume inflow from the passive model $Q_{in}^{\text{Passive}}$, respectively. Also shown is the sectional maximum SSC per semi-diurnal per tidal cycle (brown regions) and neap periods (light-gray regions).

To further investigate the cause of the variability in the TEF volume inflow between the active and passive models, we

compared the longitudinal density gradient with the difference in TEF inflows from each model ($Q_{in}^{\text{Passive-Active}}$) at each along-channel location considered. We also quantified the cross-sectionally averaged vertical density gradient including salinity and SSCs ($-\partial\rho_{all}/\partial z$), only including salinity ($-\partial\rho_{Sal}/\partial z$), and only including SSC ($-\partial\rho_{ssc}/\partial z$) (Figure 4).

The longitudinal density gradient between the models showed large ($>0.01$ kg m$^{-3}$ km$^{-1}$) differences only during spring tides. The consistent positive difference of $\partial\rho/\partial x^{\text{Passive-Active}}$ indicates that the longitudinal density gradient decreases in the

active model or when SSC are included in water density calculations. Downstream the vertical density gradient is dominated



| Section | Active & Passive (% Diff) | Neap %Diff (StdDev) | Spring %Diff (StdDev) |
|---------|---------------------------|---------------------|------------------------|
| Downstream | 3.7 (-) | 3.3 (3.1) | 6.7 (7.2) |
| ETM | 1.35 (-) | 12.5 (10.8) | 22.2 (19.2) |
| Upstream | 2.1 (+) | 73.2 (76.7) | 69.9 (61.5) |

**Table 1.** TEF bulk-values percentage differences between models. The second column (Active & Passive) shows the percentage difference between the bulk-values for the inflow in the corresponding section for the period shown in Figure 3. The neap and spring columns corresponds to the mean and the standard deviation (in parentheses) of the percentage difference of the bulk-values for the 10 neaps and spring periods shown in Figure 3. The sign indicates the effect of considering SSC into water density. A positive sign indicates that the active model increase respect to the passive model whereas a negative sign indicates that the active model decrease respect to the passive model.

by salinity and only during spring tides SSCs increases amplifying $-\partial\rho_{ssc}/\partial z$ (Figure 4b). The situation is similar in the most upstream section, but the roles of salinity and SSCs are reversed due to the lack of salinity in the water column in this section. In fact, the vertical salinity gradient is less than 0.05 kg m$^{-4}$ throughout the study period and the vertical density gradient $-\partial\rho_{all}/\partial z$ is clearly dominated by $-\partial\rho_{ssc}/\partial z$ (Figure 4d).

While the changes in TEF volume inflow are not-significant on the downstream region (Figure 4b), the overall change landward was a decrease in the TEF volume inflow when the longitudinal density gradient decreased during spring tides. During spring tides, the vertical density gradient due to sediments ($\partial\rho_{ssc}/\partial z$) was not null at the ETM and upstream. During neap tides at the ETM, the vertical density gradient was enhanced due to salinity effects when SSCs were low. The interplay between salinity and SSCs produces a shorter variability over the vertical density gradient not observed in the other locations. A

wavelet analysis was performed over 5 months of the vertical density gradient and its components at the ETM to determine the dominant frequencies of variability (Figure 5). In all three gradients, the dominant significant frequency bands were between 7-30 days. The highest energy was at 28-32 days in the vertical density gradient ($\partial\rho_{all}/\partial z$) while the salinity and SSC component have a maximum at 14-16 days. Since the impact of suspended sediments was large at the ETM, where the salinity and SSC component of the vertical density gradient showed a fortnightly variability, in the discussion we explore why these frequencies

appear.

### 4.3    Effects of SSC on stratification along-estuary

To understand the changes observed on TEF, here we examine the along-channel variability of the cross-sectionally averaged SSC, density, and Richardson number (and its components), averaged over 5 months (February-June) for both the active and passive models (Figure 6).

While the location of the ETM does not change considerably between the models (Figure 6a,f), larger SSC occurred in the case of the passive model near the surface. The locations of the isopycnals and the isohalines are similar in the downstream region of the estuary in both models (Figure 6a,b,f,g). In the upper-reaches of the estuary, the isopycnals show enhanced stratification in the active model compared to the passive model (upstream of ∼30 km, Figure 6a,f). The most pronounced





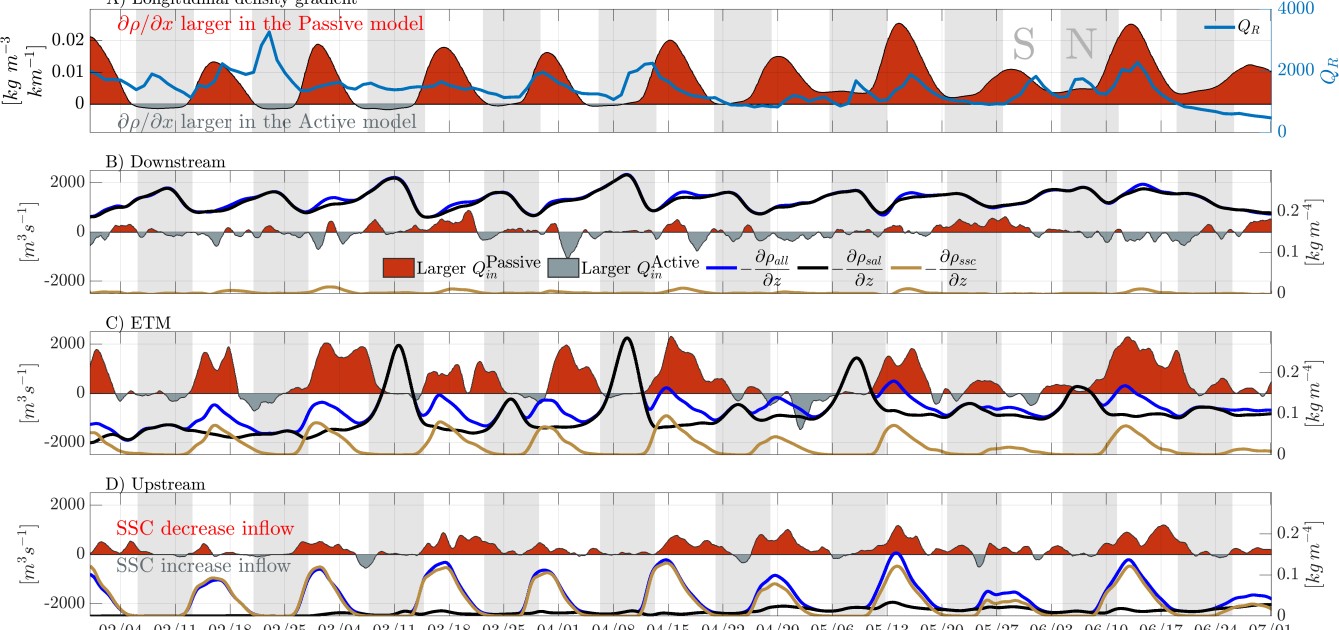

**Figure 4.** Differences in the longitudinal density gradient between models $\partial\rho/\partial x^{\text{Passive-Active}}$ and the combined river discharge are shown in (A). The red shaded region depicts when the longitudinal density gradient was larger in the passive model ($\partial\rho/\partial x^{\text{Passive-Active}} > 0$) and the gray shaded region indicates when the gradient was larger in the active model ($\partial\rho/\partial x^{\text{Passive-Active}} < 0$). Also shown are the differences on TEF volume inflow between the models ($Q_{in}^{\text{Passive-Active}}$) at the downstream (B), the ETM (C) and upstream (D) sections. The red shaded regions depicts when the inflow from the passive model is larger (SSC decrease the inflow), and the gray shaded regions depicts when the active model is larger (SSC increase the inflow). The sectional-averaged vertical gradient of the density due to both SSC and salinity ($\partial\rho_{all}/\partial z$, blue), the sectionally averaged vertical gradient of the density due to salinity ($\partial\rho_{Sal}/\partial z$, black), and sectionally averaged vertical gradient of density due to sediments ($\partial\rho_{SSC}/\partial z$, brown) are also shown. The light-gray regions depicts the neap tides periods.

difference between the two models is the enhanced buoyancy frequency in the upper-reaches of the estuary, between $\sim$30 and
$\sim$62 km, in the active model (Figure 6c,h).

    The squared vertical shear is larger throughout the estuary, albeit slightly, in the active model, indicating that the sediments enhanced vertical shear of horizontal velocity. Despite the minimal difference in $S^2$ between the models, the change in stratification quantified through $N_{all}^2$ is large enough to shift the region where $\text{Ri}_{all} > 0.25$, or where mixing is expected to be suppressed by buoyancy. In the case of the passive model, the contour line of $\text{Ri}_{Sal} = 0.25$ reached the surface at $\sim$30 km
from the mouth. In the case of the active model, this region extends landward an additional 30 km, suggesting that buoyancy driven by sediment-induced stratification is strong enough to inhibit near-surface mixing upstream of the salinity intrusion limit (Figure 6c,e,h,j).




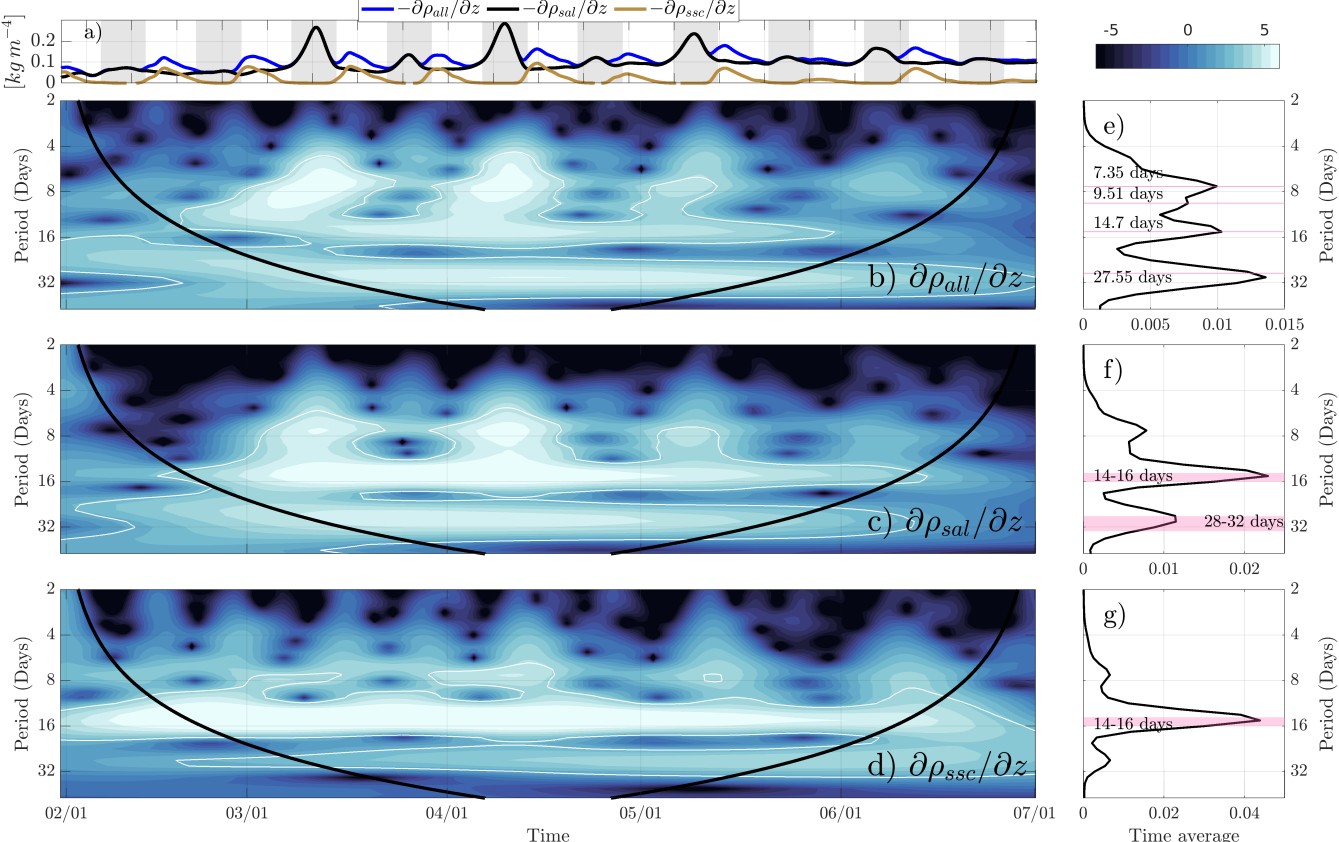

**Figure 5.** Panel (a) are the time-series of the sectional-averaged vertical gradient of the density and its components at the ETM. The wavelet analysis of the ETM sectional-averaged (b) vertical density gradient including salt and SSC ($\partial\rho_{all}/\partial z$), (c) vertical density gradient including salinity only ($\partial\rho_{Sal}/\partial z$) and (d) vertical density gradient including SSC only ($\partial\rho_{SSC}/\partial z$). On the right-hand side also shown is the time-averaged spectrum. On the wavelets, the black contour denote the cone of influence and the 95% confident interval is denoted as a thick white line.

## 5 Discussion

This study aims to determine when and where density is impacted by SSCs and how this impact influences the total exchange flow (TEF). We have found that density is affected in both its vertical and longitudinal structure, but to varying degrees along the estuary and with time. The changes in the baroclinic forcing leads to changes in TEF inflow. In this section we will first discuss the effects of SSCs on the stratification, focusing on how it varies longitudinally in the estuary and elaborating on the dominant frequencies of variability. Next, we analyze how these longitudinal and temporal changes are translated to TEF.





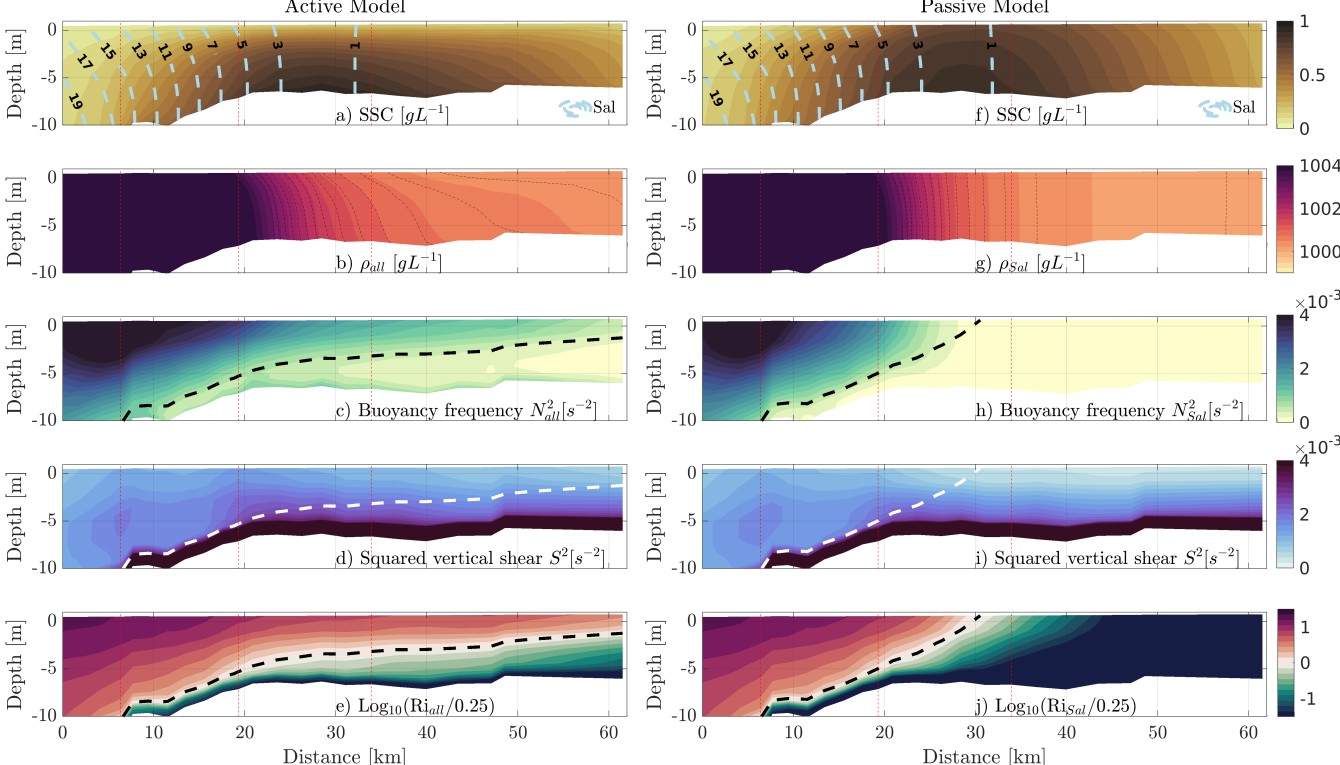

**Figure 6.** Cross-sectional and time average of (a) SSC, (b) density, $\rho$, (c) buoyancy frequency, $N^2$, (d) squared vertical shear, $S^2$ and (e) the normalized Richardson number for the active model. The plots (f)-(j) are the same as (a)-(e) but for the passive model. The averaged period corresponds to the time between the horizontal black lines in Figure 2 (February-June). In the first row, also depicted are the average salinity contours (dashed-light-blue lines). In the second row, highlighted in black are the contours of density from 1000 $gL^{-1}$ up to 1003 $gL^{-1}$ on steps of 0.2 $gL^{-1}$. The downstream region has been saturated to highlight the changes in the upstream region. The Ri=0.25 line is highlighted in the third, fourth and fifth rows. Vertical red lines represent the location of the sections from Figure 2.

## 5.1 Fine sediment effects on stratification

This study confirmed previous studies showing that SSCs induce effects over density and stratification (Figure 6). The buoyancy frequency and gradient Richardson number aided in answering the first research question of this study: where is density impacted by SSC? The upstream stretch show the largest differences between the active and passive models, with a stronger density gradient in the active model that increases both buoyancy frequency and gradient Richardson number. This indicates that buoyancy dominates over shear production in this upstream stretch when sediments are considered in the quantification of

water density.

Using an one-dimensional vertical (1DV) numerical model, Winterwerp (2001) argued that the main contribution of large SSCs to estuary hydrodynamics was to modify the velocity shear, which in turn changes stratification. However, while we





found that the incorporation of SSCs in water density affects the velocity shear (Figure 6d,i), these changes were small: the differences were of the order of $\mathcal{O}(1 \times 10^{-4}\ \mathrm{s}^{-2})$ between the models. We find that the primary effect of SSCs was to enhance

stratification causing changes of the order of $\mathcal{O}(1 \times 10^{-3}\ \mathrm{s}^{-2})$ (Figure 6c,h and e,j). The elevated stratification caused by SSCs in the active model, inhibits the vertical flux of SSC, allowing a larger accumulation of sediments near the bed (Figure 6a). This explains the strengthening of the vertical gradient of SSCs near the ETM in the active model (compare Figure 6a,f). These results are in agreement with Zhu et al. (2021) who also found that the vertical sediment flux decreased due the sediment-induced density gradient. Further, the sediment-induced stratification alters settling velocities which play a major role in the

maintenance of the ETM (van Maanen and Sottolichio, 2018). These effects increase near the ETM due to the proximity of the salinity intrusion limit and large SSCs (Zhu et al., 2021) as well as flocculation or hindered settling (Dijkstra et al., 2018). In addition, sediment-induced stratification can alter the shape of the ETM (Zhu et al., 2021) with a larger extension upstream compared to downstream as was found in our model (Figure 6a compared to f). Because the extension of the ETM is larger when sediments are included in water density, the region where SSCs dampen shear-induced mixing is extended upstream as

well, explaining the larger region where the Ri >0.25 (Figure 6e,j).

    Zhu et al. (2021) found that sediment-induced density gradients are pronounced only when salinity stratification is also present. In our case, the effects of stratification caused by SSCs were more relevant in regions where they correspond to a large fraction of the total stratification, but did not completely dominate, suggesting that the salinity stratification is also important. At the downstream location, the vertical density gradient ($\partial \rho_{all}/\partial z$) is largely dominated by salinity ($\partial \rho_{Sal}/\partial z$)

as the vertical density gradient with sediments ($\partial \rho_{ssc}/\partial z$) is negligible except for small increases during spring tides (Figure 4b). The upstream location is close to the salinity intrusion limit (Figure 2c,d). Because of this, the vertical density gradient computed only with salt ($\partial \rho_{Sal}/\partial z$) was small in this section (Figure 4d) allowing the SSCs to play a larger role in suppressing mixing (Figure 6).

    At the ETM we found that salinity and SSCs enhanced the total vertical density gradient ($\partial \rho_{all}/\partial z$) at different phases of the

spring-neap cycle. The magnitude of $\partial \rho_{Sal}/\partial z$ and $\partial \rho_{ssc}/\partial z$ were of similar order, except for three neap tides when $\partial \rho_{Sal}/\partial z$ exceeded $\partial \rho_{ssc}/\partial z$ (March 11, April 12, May 9; Figure 4b), but featured different fortnightly variability. The weaker currents during neap tides affected density in two ways: (1) they allow the settlement of suspended sediments decreasing the vertical SSC gradient and (2) the vertical salinity gradient increases due to less intense mixing. These two effects cause the salinity to dominate the $\partial \rho_{all}/\partial z$ at the ETM during neap tides. Similarly, there are two effects on density during spring tides: (1)

the resuspension of sediments intensify the vertical SSC gradient and (2) salinity-induced stratification decreases because of enhanced turbulent mixing. This enhances $\partial \rho_{ssc}/\partial z$ and decreases $\partial \rho_{Sal}/\partial z$ during spring tides, and both are modulated on the fortnightly cycle, but on opposite phases (Figure 4c). It has to be noted that the model did not reproduce benthic fluid mud layers, where SSC can reach 100 g L$^{-1}$ over 1 m thick. These layers form typically in neap tides, and can significantly affect near-bed turbulence due to high local density stratification and increased water viscosity (Le Hir et al., 2000).

To extract the dominant periods of variability, we applied the wavelet analysis to each of the three vertical density gradients (Figure 5). The vertical density gradient with only salinity ($\partial \rho_{Sal}/\partial z$) shows a dominant fortnightly periodicity with a lesser lunar monthly (28-32 d) periodicity. The vertical density gradient with only SSCs ($\partial \rho_{ssc}/\partial z$) also shows a dominant fortnightly



periodicity, with very little energy at other periodicities. However, the total vertical density gradient ($\partial\rho_{all}/\partial z$) shows several additional dominant periods. The modulation of the fortnightly cycle of the $\partial\rho_{Sal}/\partial z$ and $\partial\rho_{ssc}/\partial z$, interact to produce a dominant variability of 7.35 d in $\partial\rho_{all}/\partial z$. The fortnightly variability of the vertical SSC gradient (14.7 d) and the monthly lunar variability of the salinity (27.55 d) also interact to create a distortion (subtraction) of 9.51 d and a modulation (sum) of 31.84 d on $\partial\rho_{all}/\partial z$ (Figure 5e), just like compound tides (Valle-Levinson, 2022). Because of these interactions the band of 28-32 d is the most energetic band for the total vertical density gradient ($\partial\rho_{all}/\partial z$). An analysis on the passive model revealed that this frequency band is not present when SSCs are not considered in water density calculations (not shown). These results highlight that neglecting SSCs in the quantification of water density will exclude potentially influential dynamics affecting stratification in estuaries, in particular close to the ETM.

## 5.2 Suspended sediment effects on exchange flows

We have found that the omission of suspended sediments in the calculation of water density leads to differences in TEF values (Figure 3 and Table 1). The percent difference of the bulk-values from the active to passive model when quantified over the entire study period (5-months) were found to be relatively small (less than 4%), likely because of the enhancement of SSCs during only the spring tides, allowing for their contribution to be diminished when averaged over long time-periods. This was reinforced by the quantification of TEF bulk values during neap and spring tides separately, which produced percent differences between the active and passive models that were considerably larger, specifically at the upstream location (up to 70%).

Using an analytical model, Talke et al. (2009) showed that the sediment-induced density gradient enhances the salinity-driven estuarine circulation upstream of the ETM whereas it causes a decrease downstream of the ETM. Similar results were later confirmed by a 3D modeling study in the Yangtze estuary (Zhu et al., 2021). Since we are not considering estuarine circulation but rather calculations of the inflow volume transport using salinity coordinates (TEF), we can not directly compare results with those of Talke et al. (2009), but some linkages can be made. Our results show that the most significant impact of SSCs on the TEF volume inflow were observed at the ETM and at the upstream location. Although slightly, the bulk-values over the entire period agree with both papers as the TEF volume inflow decrease at the downstream and the ETM locations whereas increases at the upstream location (Table 1). However, SSC-induced effects were not in agreement with either of the above mentioned studies when considering tidal bulk-values for neap and spring. Both of these show a decrease in the active model indicating that the SSCs induced effect was to decrease the TEF volume inflows on tidal scales. In particular, at the ETM and the most landward location, the SSCs tended to decrease the TEF inflow ($Q_{in}^{\text{Passive-Active}} > 0$) rather than increase it (Figure 4b,c). The most pronounced changes in $Q_{in}^{\text{Passive-Active}}$ occurred during spring tides at the ETM and upstream when SSCs reached concentrations of ~6 and ~4 g L$^{-1}$, respectively. At the downstream location $Q_{in}^{\text{Passive-Active}}$ was overall smaller than at the ETM and upstream locations, even during spring tides, likely due to the location being far enough from the ETM that $\partial\rho_{ssc}/\partial z$ was negligible. The changes in the longitudinal density gradient are likely the cause of the temporal modulation of $Q_{in}^{\text{Passive-Active}}$.

Figure 7 schematizes the changes due to SSC. The incorporation of SSC in water density calculations increases the stratification in the upper-reaches of the estuary. Despite these changes not being large, they decrease the depth-averaged longitudinal




density gradient, which in turn allows for less intense exchange flows than when sediments are not included in density calculations (Figure 7). The changes in the fortnightly variability of $\partial\rho_{ssc}/\partial z$ and $\partial\rho_{sal}/\partial z$ at the ETM produces a near weekly modulation of the vertical density gradient (Figure 4). Both the vertical and horizontal dynamics will ultimately be neglected
when SSCs are not incorporated into water density calculations, causing overestimates of the exchange inflow $Q_{in}$ leading to incorrect assumptions about systems.

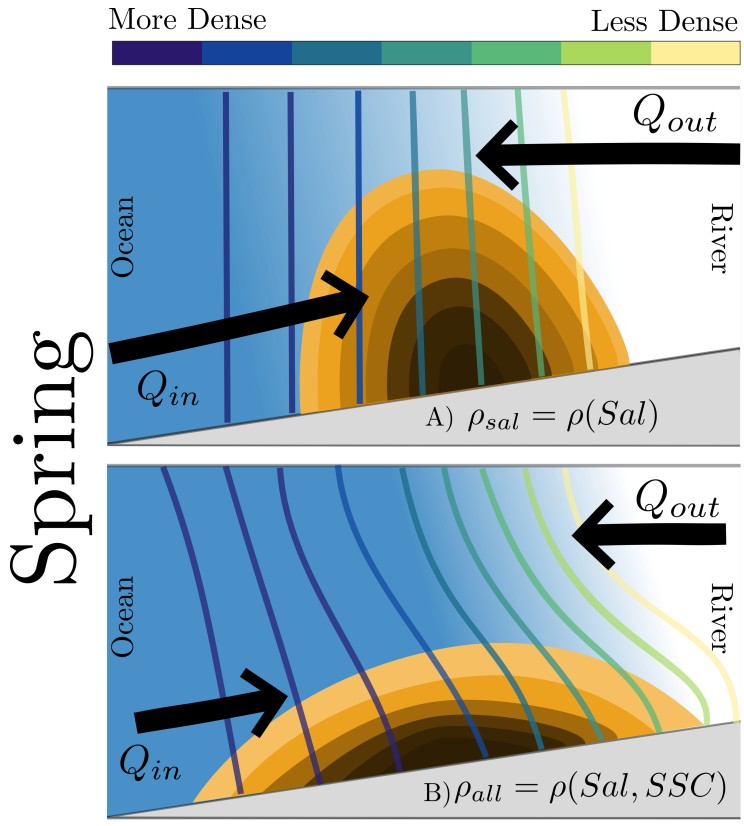

**Figure 7.** Schematized changes in the longitudinal density gradient due to the consideration of suspended sediment concentration on water density. The vertical lines depicts isopycnals from saltwater to freshwater (colorcode) while arrows size depicts magnitude of the exchange flow.

### 5.3 Model caveats

This study was designed to analyze the contribution of SSCs to water density and the TEF in a macrotidal and turbid estuary. To accomplish this goal several assumptions and simplifications were made. It is important to note that sediment dynamics are very
sensitive to the parametrization of settling velocity. Van Leussen flocculation formulation has proven effective in reproducing the position of the ETM, with fairly realistic SSC on spring tides but with too low values during neaps tides. Vertical gradients and effect of near-bed SSC on stratification were thus underestimated during neaps.



More research is needed to determine the extent of the present results with consideration of different particle sizes, settling velocities, and sediment concentrations. As every parameterization, the van Leussen parameterization used in this study has advantages and limitations for reproducing SSC observations and ETM dynamics (Do et al., 2025). While this formulation has clear advantages for reproducing the upstream migration of the ETM and capturing high SSC levels (Do et al., 2025), it also tends to underestimate SSC near the bottom during neap tides which could likely affect vertical fluxes. The extension of our results to other systems needs that the model is able to reproduce to right orders of magnitude of SSC in each system, and thus more careful calibration of settling velocity is recommended.

Also relevant in this study is the consideration of non-linear effects between salinity and SSCs when calculating water density. In this study, we followed the previously employed assumption that salinity and suspended sediments can be added linearly to the water density calculations used by several authors (i.e., Brenon and Le Hir, 1999; van Maanen and Sottolichio, 2018; Talke et al., 2009; Guan et al., 2005). In the study by Guan et al. (2005), for example, the authors include non-linear effects between salinity and SSC (their Eq. 7), but they implicitly work on a linear relationship ignoring the non-linear effects of salinity and SSCs. This was likely because the contribution of the non-linear effects was regarded as small relative to the linear contributions. More recently, Zhou et al. (2025) estimated the non-linear interactions between salinity and SSC using numerical simulations. Among their findings, the authors found that the total stratification resulting from salinity and SSC was significantly different than the sum of the individual effects. Further, they also identify that the non-linear interaction between the salinity-induced stratification and the sediment-induced stratification vary from stabilizing to destabilizing depending on the tidal phase. Despite the limitations of the study as a one-dimensional numerical model, the findings by Zhou et al. (2025) reveal that the consideration of non-linear effects between salinity and SSC can be important in the study of suspended sediments and thus in estuarine exchange flow studies as the flood-ebb asymmetry could also cause exchange flows similarly to the SIPS (Simpson et al., 1990).

Also, the use of TEF in hyperturbid systems must be interpreted carefully. Because of the use of salinity coordinates, the spatial variability of the volume transports (the TEF inflow and outflow) can only be assumed in relation to where one would expect salinity classes to be (i.e., where water is brackish versus the upstream riverine section that may still be tidal). However, hyperturbid systems could present stratification where there is no salinity. Further, a consequence of the sediment-induced stratification is the existence of inverted salinity profiles. These have been reported in systems with large SSCs such as the Gironde Estuary (Sottolichio et al., 2011), the Ems estuary (Becker et al., 2018), the Ouse estuary (Uncles et al., 2006) or the Huanghe (Wang and Wang, 2010). The existence of these inverse salinity profiles challenges the assumption that higher salinity values must be closer to the bottom advising caution on how to interpret TEF profiles in hyperturbid systems. As a final remark, it is important to highlight that this study was focused on high-river regime conditions. The conditions that exist during low river regime will likely modify the dynamics along the estuary and therefore warrants further investigation.



## 6 Conclusions

This study aimed to determine when and where water density, and subsequently estuarine exchange flows, are impacted by SSCs in a macrotidal and turbid estuary. Is the first time a study analyze the effects of suspended sediments concentrations over the total exchange flow methodology. Using semi-realistic numerical simulations, we found that downstream of the ETM, SSCs contributed little to the water density causing small changes on density and only during spring tides. Thus, during spring tides downstream of the ETM, both stratification and TEF were not largely affected. At the ETM, the importance of SSCs

increase and has a more dominant role than salinity during spring tides whereas salinity still dominates during neap tides. The fortnightly variability of both SSCs and salinity affect the vertical density gradient and the modulation and distortion of the $\partial\rho_{Sal}/\partial z$ and $\partial\rho_{ssc}/\partial z$, produce dominant variabilities of 7.35 d, 9.51 d and 31.84 d on $\partial\rho_{all}/\partial z$, just like compound tides. Ultimately, this dynamic will be neglected when SSCs are not incorporated into water density calculations. We found that SSCs effects are even more important upstream of the ETM where they dominate the stratification which cause inhibit of mixing in

a region of $\sim$30 km.

The change in TEF bulk-values for the entire period (5 months) are in agreement with previous results that showed that the estuarine subtidal circulation decreases downstream of the ETM but increases upstream when SSCs are incorporated in water density calculations. However, tidal bulk-values quantified only during neap or spring tide indicate a decrease in all sections. Although the longitudinal density gradients produced solely by SSCs are small, not considering the contribution of SSCs to

445 water density calculations in this region could lead to incorrect assumptions on exchange flow. The increased stratification caused by SSC acts to decrease the baroclinic forcing of the exchange flow. Not considering SSCs in water density may lead to overestimates of TEF as high as 70% during spring tides. Further research is needed with more realistic simulations of ETM and to determine if this phenomenon continues as the ETM migrates along the estuary (seasonal or inter-annual comparisons) and to confirm this finding in other highly turbid estuaries.

*Data availability.* The raw data supporting the conclusions of this article will be made available by the authors.

*Author contributions.* CMR: Conceptualization, Methodology, Analyses, Writing  original draft, Data curation, Writing  review & editing. LR: Conceptualization, Writing - original draft, Writing  review & editing, Supervision, Funding. BJK: Methodology, Data curation, Software. IJR: Writing - review & editing. AS: Writing  review & editing. NH: Data curation, Software.

*Competing interests.* The authors declare that they have no conflict of interest



*Acknowledgements.*  Cristian M. Rojas and Lauren Ross were funded by the National Science Foundation Grant No. 2045866. This work was supported in part through the computational resources and staff expertise provided by Advanced Research Computing, Security, and Information Management (ARCSIM) at the University of Maine at Orono.



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
