# Peer review of "The impact of suspended sediments on exchange flow in a macrotidal, hyperturbid estuary."

_EGUsphere, 2025_

## Referee Comment (RC1)

**Review of Paper #egusphere-2025-6311**

**Summary and General Assessment**

This manuscript examines how including Suspended Sediment Concentration (SSC) in water density calculations affects the Total Exchange Flow (TEF). The authors employ a numerical model of the Gironde Estuary to compare an 'active' approach (where density depends on both salinity and SSC) with a 'passive' approach (where density depends solely on salinity). Their findings show that, although overall inflow differences between the two models are minor over seasonal periods (less than 4%), incorporating SSC can substantially modify TEF during spring tides and in the estuary's upper reaches—sometimes changing inflow estimates by up to 70%.

The study tackles an important issue in estuarine physics: how sediment dynamics interact with the TEF framework. The reported suppression of shear production and variations in stratification frequency are particularly noteworthy. I recommend a minor revision to ensure the robustness of the results. My comments below address aspects of the methodology and physical interpretation in the manuscript.

**Major Comments**

**Vertical resolution** The model uses nine equidistant terrain-following layers. The authors then compute Ri, vertical shear, $N^2$, and claim a $\sim 30$ km shift in mixing suppression. With 9 layers, shear and $\frac{\partial \rho}{\partial z}$ are very discretized, especially near-bed where SSC gradients are sharp. I would suggest at least adding a discussion of how layer count influences Ri exceedance frequency and the $\sim 30$ km claim (even if the authors can't rerun, they can sanity-check with vertical-gradient smoothing/finite-difference choices).

**Linear Superposition of Density Components** The authors calculate total density by linearly adding the effects of salinity ($\rho_{Sal}$) and suspended sediments ($\rho_{SSC}$). While the authors acknowledge that non-linear effects exist and cite recent work on the subject, the manuscript would benefit from a more quantitative justification for this simplification within the specific context of the Gironde. Given the hyperturbid nature of the system, can the authors provide an estimate of the potential error magnitude introduced by neglecting non-linear equation of state interactions?

**Minor Comments**

Page 1, Line 9: The 'occur' should be corrected to 'occurs'.

Page 3, Line 59 (and other places): The unit 'gL$^{-1}$' should be corrected to 'g L$^{-1}$' (with a space).

Page 3, Line 85: The sentence 'Is $\sim 75$ km funnel shape long...' is grammatically incorrect. Please revise to 'It is a $\sim 75$ km long, funnel-shaped estuary...'.

Page 5, Line 96: Add a space after 'sediment transport dynamics'.

Page 6, Equation 1 and 2 (and other places): Please unify the case of 'Sal' in $\rho_{Sal}$.

Page 8, Line 207: 'it well reproduced' should be corrected to 'it was well reproduced'.

Page 9, caption of Figure 2: Add 'dashed' in 'The horizontal red line'.

Page 12, caption of Table 1: 'increase respect to' should be corrected to 'increase relative to'.

Page 12, Line 273: kg m$^{-4}$ should be corrected to g kg$^{-1}$ m$^{-1}$.

Page 12, Line 275: 'not-significant' should be corrected to 'not significant'.

Page 15, Figure 6: The units in Panel (b) should be corrected to kg m$^{-3}$.

Page 20, Line 431: The sentence 'Is the first time a study analyze...' should be corrected to 'This is the first time a study analyzes...' or 'This is the first study to analyze...'.

Page 20, Line 440: The phrase 'cause inhibit of mixing' should be corrected to 'inhibit mixing' or 'cause the inhibition of mixing'.